# Visual Outcomes of Small-Incision Lenticule Extraction (SMILE) in Thin Corneas

**DOI:** 10.3390/jcm11144162

**Published:** 2022-07-18

**Authors:** Erika Eskina, Olga Klokova, Roman Damashauskas, Karine Davtyan, Bojan Pajic, Marina Movsesian

**Affiliations:** 1Ophthalmological Clinic “Sphere”, 117628 Moscow, Russia; marina.movsesyan@sfe.ru; 2Academy of Postgraduate Education of Federal State Budgetary Foundation Federal Research-Clinical Center Federal Medical-Biological Agency of Russia, 125310 Moscow, Russia; 3Krasnodar Branch of The Sviatoslav Fyodorov Eye Microsurgery Federal State Institution, 350012 Krasnodar, Russia; oaklokova@gmail.com (O.K.); quas87@gmail.com (R.D.); 4“Clear View” Clinic, 121467 Moscow, Russia; info@cvclinic.ru; 5Swiss Eye Research Foundation, Titlisstrasse 44, 5734 Reinach, Switzerland; bojan.pajic@orasis.ch; 6Eye Clinic ORASIS, Titlisstrasse 44, 5734 Reinach, Switzerland; 7Division of Ophthalmology, Department of Clinical Neurosciences, Geneva University Hospitals, 1205 Geneva, Switzerland; 8Faculty of Medicine, University of Geneva, 1205 Geneva, Switzerland; 9Faculty of Sciences, Department of Physics, University of Novi Sad, Trg Dositeja Obradovica 4, 21102 Novi Sad, Serbia; 10Faculty of Medicine of the Military Medical Academy, University of Defense, 11000 Belgrade, Serbia

**Keywords:** refractive surgery, SMILE, thin cornea

## Abstract

We aimed to find out whether thin (≤500 μm) or normal (>500 μm, control) corneal thickness would impact efficacy and safety outcomes of small-incision lenticule extraction (SMILE). We retrospectively analyzed medical records of adult patients who had undergone SMILE. A total of 57 eyes were included in the “thin corneas” group and 180 eyes in the “control” group. At one month after surgery, rates of patients with uncorrected distance visual activity (UDVA) ≥ 0.8 were significantly higher in patients from the control group compared to the “thin corneas” group (87 vs. 71%, respectively *p* < 0.01), though rates were comparable at 3 months (87 vs. 76%, respectively, *p* > 0.05). SMILE had comparable safety in patients with thin and normal corneas. Procedure result predictability was comparable between groups. Regression analysis demonstrated that cap thickness impacted posterior corneal biomechanics, and the volume of removed tissue had a higher influence in patients with thin corneas. Moreover, an increase in cap thickness was associated with better final BCVA. Further study is needed for the evaluation of the impact of thin corneas on SMILE outcomes and planning. Our study also indicates that patients with thin corneas might require a different approach to nomogram calculation.

## 1. Introduction

Achieving an extremely thin cornea was always the barrier to high myopia or myopic astigmatism correction using laser techniques [1]. On the one hand, mathematical modelling studies demonstrated that the use of lenticular surgery methods such as femtosecond lenticule extraction (FLEx) or small-incision lenticule extraction (SMILE) may spare an additional 100 μm of corneal thickness (in case of 500 μm cornea) with impact on the biomechanical strength of the cornea comparable to the effect of flap-type surgery (laser in situ keratomileuses [LASIK]) [2,3]. On the other hand, results from a tensile strength study indicate that corneal thickness has a strong influence on the biomechanical stability of the cornea [4]. Moreover, there are still questions as to whether it is only the thickness of the removed cornea that impacts corneal biomechanics, or whether the type of corneal excision also plays a significant role in the biomechanical properties of the postoperative cornea [5,6,7]. Recent studies confirmed that risk of iatrogenic corneal ectasia after any type of refractive surgery remains an unsolved issue [8].

It is widely known that the posterior corneal surface is very often a starting point for the ectatic process [9]. For that reason, prognostic indexes were developed for posterior cornea evaluation to calculate the risk of ectasia progression. Nowadays, there are a few known scales used by refractive surgeons to detect the risk of corneal ectasia, such as the Belin/Ambrosio enhanced ectasia total derivation (BAD-D) [10], the Keratoconus Prediction Index (KPI) and the inferior–superior value (ISV) [11]. It was also shown that a combination of some special high-order aberration (HOA) assessment methods could illustrate the risk and progression of keratoconus [9,12]. For instance, the combined Scheimpflug–Placido unit “Sirius” (Costruzione Strumenti Oftalmici, Florence, Italy) allows obtaining several indexes:Symmetry index front (Sif) and symmetry index back (SIb);Keratoconus Vertex front (KVf) and Keratoconus Vertex back (KVb);Baiocchi–Calossi–Versaci front (BCVf) and Baiocchi–Calossi–Versaci front and back (BCVb).

All these indexes allow the eye practitioner to successfully distinguish the subclinical keratoconus from normal corneas [13].

In the clinical setting, thin corneas are always challenging decision making towards better ways of refractive error correction and are associated with a higher risk of postoperative ectasia [14]. It is also known that posterior cornea curvature is changing as a result of the refractive lenticule extraction (ReLEx) SMILE procedure, demonstrating the biomechanical impact of this surgery method [15].

There are still many unsolved questions regarding laser refractive surgery, including the possibility of residual stromal thickness (RST) and tissue removal extension while performing SMILE. The other controversial topic is the use of lenticular surgery in patients with an expected RST of less than 300 μm. For example, the use of preventive corneal crosslinking in patients with an RST of 300 μm or less was only partially effective in terms of functional outcome [16]. 

Our hypothesis was that thin corneas could be as good a biomechanical model to find the limits of RST and tissue removal as average-thickness corneas. In the present study, we aimed to study the reaction of the posterior corneal surface in patients requiring high refractive error correction using the ReLEx SMILE approach.

## 2. Materials and Methods

### 2.1. Study Setting

This was a retrospective multicenter study with participation of two eye surgery clinical centers: Ophthalmological Clinic “Sphere” (Moscow, Russia) and S.N. Fedorov NMRC “MNTK “Eye Microsurgery” (Krasnodar, Russia). Study was approved by SPHERE Eye surgery Clinic institutional review board of ethics committee.

SMILE was performed in adult patients aged 18–44 years with low to high myopia. According to the surgical protocol of both centers, minimum lenticule thickness was programmed to 15 μm. Pre-planned RST was set > 290 μm. Patients were followed up until 6 months post-surgery.

### 2.2. Data Collection

Patient data were collected from source medical documents available in the participating centers. We used deidentified data and considered data collection as secondary. We collected data on pre- and post-surgery visual acuity, corneal pachymetry, refraction dynamics, the predictability of procedure results, surgery plan, as well as topographic measurements and Scheimpflug analysis, including KVb, μm (maximum posterior elevation); Radius flat (Rf) back and Radius steep (Rs) back at 6 mm diameter of the cornea; root mean square per area (RMS/A) of the posterior corneal surface.

Sif and Sib were used as measures of vertical asymmetry: positive values indicated the inferior hemisphere was steeper than the superior, negative values indicated the superior hemisphere was steeper than the inferior.

KVf and KVb indicated the highest point of ectasia on the anterior and posterior elevation maps. BCVf and BCVb were used to evaluate presence of ectasia through analysis of coma and trefoil components of Zernike’s decomposition of elevations in zones where keratoconus arises.

At surgery plan evaluation, attention was paid to the following parameters:Optical zone size (OZ) chosen for the SMILE procedure; cap diameter (D-cap); RST; lenticule thickness; the volume of removed tissue.

Patients with missing data were not included in this study. 

### 2.3. Statistics

For the purposes of the current study, treated eyes were divided into two cohorts (strata). Eyes with thin corneas, which had a baseline minimal corneal thickness of ≤500 μm are further addressed as the “study group” or the “thin corneas” group. Eyes with a baseline corneal thickness of >500 μm are addressed as the “control” group. 

Statistical analysis was performed using R version 3.6.3. Descriptive statistics were presented as numbers of patients (percent), means and standard deviations (SD), and ranges (where applicable). Quantitative variables were analyzed using the Mann–Whitney U test or the Wilcoxon signed-rank test. The Mann–Whitney test was chosen due to the rejection of normality hypothesis for continuous variables checked by the Shapiro–Wilk test. Multilinear regression analysis was conducted starting with full degrees of freedom and applied stepwise regression. Additionally, ordinary least square method and coefficient of determination (R^2^) were used to support the analysis. Differences were considered statistically significant at *p*-level < 0.05.

## 3. Results

### 3.1. Population Characteristics

A total of 119 patients (237 eyes) were identified as applicable for current study. Baseline (pre-surgery) population characteristics are outlined in Table 1. 

The thin corneas group included 29 patients (57 eyes) with a mean age of 31.47 ± 7.21 years, a mean sphere of −4.41 ± 2.14 D, ranging from −9.5 to −1.5 D, and a corneal thickness of 487.77 ± 9.01 μm (range 460–500). The control group included 90 patients (180 eyes) aged 29.71 ± 6.91 years, a mean sphere of −4.76 ± 2.17 D, ranging from −11 to −1.25 D, and a corneal thickness of 543.06 ± 30.99 (range 502–641).

There were no statistical differences regarding baseline parameters including age, sphere and cylinder, spherical equivalents (SEQ) and best corrected visual acuity (BCVA) (*p* > 0.05 for all intergroup comparisons). Corneal thickness was larger in the control group (*p* > 0.05). Considering the difference in baseline corneal thickness, it was expected that observed surgery parameters such as OZ, RST, D-cap, lenticule thickness, and the volume of removed tissue would differ (*p* < 0.001 for all intergroup comparisons). Target refraction was lower (in some cases, myopic because of tissue limit) in patients with thin corneas (*p* = 0.034 for intergroup comparison).

### 3.2. Efficacy

Figure 1 demonstrates uncorrected distance visual activity (UDVA) recovery 1 and 3 months after the SMILE procedure. We found that in patients from the thin corneas group one month after surgery, the percent of eyes with UDVA ≥ 0.8 was significantly higher in eyes from the control group compared to those from the “thin corneas” group (87 vs. 71%, respectively *p* < 0.01). Three months after SMILE, the difference was not statistically significant (87 vs. 76%, respectively, *p* > 0.05). At six months, more eyes with thin cornea had UDVA ≥ 0.8 compared to controls (98 vs. 91%, respectively, *p* = 0.003). Of note, at six months, data were slightly limited for both groups since many patients skipped the appointment, with a total of 46 eyes (180 at baseline) from the control group and 17 eyes (57 at baseline) from the “thin corneas” group with evaluable UDVA measurements. Therefore, data at six months should be interpreted cautiously. 

We also compared the difference between baseline BCVA and postoperative UDVA. There was a statistically significant intergroup difference between preoperative BCVA and postoperative UDVA at 1 month—64 and 50% of eyes from the control and “thin corneas” groups, respectively, had the same or a better result (*p* < 0.01)—while results were comparable at 6 months (*p* > 0.05), indicating slightly slower recovery in patients with thin corneas. 

### 3.3. Safety

Due to the retrospective nature of this study, we evaluated the change in BCVA after surgery to compare the safety profile depending on baseline cornea thickness (Figure 2). At 3 months (Figure 2A) after surgery, 93% of eyes in patients from the control group and 86% of eyes in patients from the “thin corneas” group had the same or better BCVA (*p* > 0.05). At 6 months after the procedure (Figure 2B), 91% of eyes from the control group and 95% of eyes from the “thin corneas” group had the same or better BVCA (*p* = 0.08). It is also noteworthy that at 6 months after surgery, SMILE appears to be safe according to refractive surgery standards (e.g., comparable number of eyes lost >1 Snellen of BCVA).

Mean BCVA values were significantly lower in eyes with thin corneas than in eyes from the control group: 0.93 ± 0.19 vs. 1.01 ± 0.11 D and 0.92 ± 0.11 vs. 1.01 ± 0.09 D, respectively (*p* < 0.05 for both comparisons). There was no significant difference in mean BCVA at 6 months (0.99 ± 0.03 vs. 0.99 ± 0.09, *p* > 0.05). Therefore, slower recovery of visual acuity was not a sign of inferior SMILE safety in patients with relatively thin corneas. 

### 3.4. The Predictability of Results

To evaluate the predictability of surgery results, we performed correlation analysis for planned and achieved spherical equivalent (Figure 3). There was a strong linear correlation between attempted and achieved spherical equivalent at 3 months (R^2^ = 0.9842 for the control group and 0.8832 for the “thin corneas” group) and at 6 months. We can assume a slower stabilization of the refractive effect in patients with thin corneas, which could be what led to vision acuity recovery in those patients.

We found no significant difference in spherical equivalent refractive accuracy at 6 months (Figure 4).

One month after the procedure, spherical equivalent stability (Figure 5) was −0.45 ± 1.01 D in the eyes of patients from the “thin corneas group” and −0.11 ± 0.45 D in the eyes of patients from the control group (*p* < 0.05). At 3 months, spherical equivalent stability was −0.42 ± 1.03 and −0.09 ± 0.33 D in patients from the “thin corneas” and control groups, respectively (*p* > 0.05). There were no significant differences in spherical equivalent at 6 months (*p* > 0.05). 

### 3.5. Scheimpflug Analysis

We evaluated KVb as a measure of the apex of posterior elevation. Preoperative KVb was 9.73 ± 2.53 μm in patients from the control group and 7.96 ± 2.14 μm in patients from the “thin corneas” group (*p* < 0.001). At 3 months after surgery, KVb was 11.45 ± 5.98 μm and 8.09 ± 3.48 μm in patients from the control and “thin corneas” groups, respectively (*p* > 0.05). However, there was no intergroup difference in postoperative KVb change (*p* > 0.05).

Rf back at 6 mm before surgery was −6.03 ± 0.26 and −6.09 ± 0.33 in patients from the control and “thin corneas” groups (*p* > 0.05). At 3 months, Rf back was −6.22 ± 0.27 and −6.38 ± 0.25, respectively (*p* < 0.05). We found no significant difference in Rs back before and after surgery (*p* > 0.05 for all intergroup comparisons). 

Preoperative RMS/A was 0.07 ± 0.02 in patients from the control group and 0.08 ± 0.02 in patients from the “thin corneas” group (*p* > 0.05). At 3 months after the procedure, RMS/A was 0.13 ± 0.19 in patients from the control group and 0.11 ± 0.03 in patients from the “thin corneas” group (*p* > 0.05). 

### 3.6. Factors Impacting Efficacy and Biomechanical Behavior of the Cornea

Regression analysis led us to several insights into factors impacting the efficacy and the predictability of SMILE in patients with thin corneas. 

#### 3.6.1. Rf Back and Rs Back

In patients from the control group, an increase in the volume of removed tissue by 1 mm^3^ would lead to a decrease in Rs back by 0.085 at 3 months postop. Therefore, an increase in the volume of removed tissue in patients with normal corneal thickness will lead to a smaller change in Rs back. On the contrary, in patients with thin corneas, a volume of 1 mm^3^ of removed tissue would increase Rs back by 0.149 compared to the control group. In patients with thin corneas, the volume of removed tissue had a greater impact on corneal biomechanics. 

In patients from the control group, an increase in the volume of removed tissue by 1 mm^3^ would lead to a decrease in Rf back by 0.077 at 3 months. Therefore, an increase in the volume of removed tissue in patients with normal corneal thickness will lead to a smaller change in Rf back. In the control group, a cap thickness of 120 μm was associated with a greater (by 0.148) change in Rf back, than in patients with a cap thickness of 110 μm. No significant correlations were found for Rf back in patients from the thin corneas group.

In the control group, a cap thickness of 120 μm was associated with a greater (by 0.102) change in Rs back, than in patients with a cap thickness of 110 μm. No significant correlations were found for cap thickness and Rs back in patients from the thin corneas group. 

Therefore, cap thickness impacts the posterior corneal surface and might influence the biomechanics of the cornea on SMILE surgery. 

#### 3.6.2. RMS/A

A cap thickness of 120 μm in patients from the control group was associated with a 0.112 lower change in RMS/A compared to patients with a cap thickness of 110 μm. No significant correlations were found for RMS/A in patients from the thin corneas group.

#### 3.6.3. BCVA at 3 Months

A cap thickness of 120 μm in patients from the control group was associated with a 0.044 lower change in BCVA compared to patients with a cap thickness of 110 μm. No significant correlations were found for BCVA in patients from the thin corneas group. Therefore, our data indicate that in patients with a good RST, a higher cap thickness could boost vision recovery, while safety concerns should be considered first in patients with thin corneas and/or a high volume of altered tissue.

## 4. Discussion

There are several studies reporting the efficacy and the safety of SMILE in patients with different corneal thicknesses (Table 2). Kabakci et al. (2020) published a report covering 55 eyes of 39 patients who had a preoperative central corneal thickness of 470–485 or 485–500 μm. The authors reported the difference in postoperative RST, KVb, or corneal thickness and concluded that the procedure was safe for patients regardless of corneal thickness. At the same time, at 24 months follow up, higher rates of HOA were reported. It should also be noted that a corneal thickness of < 475 μm is among the contraindications for SMILE surgery [17]. Ganesh et al. (2015) published an efficacy and safety study of SMILE combined with collagen crosslinking in patients with thin corneas (mean central corneal thickness of 501.3 ± 25.9 μm) and borderline corneal topography (*n* = 40 eyes). The authors reported good efficacy, with postoperative UCVA of 20/25 or better in all eyes. At 12 months after surgery, there were no cases of haze, keratitis, ectasia, or regression [18]. A recently published study by Zhao et al. (2022) describes a 3-year follow up of patients with thin corneas who had undergone SMILE (*n* = 97 eyes in 97 patients). The authors evaluated outcomes in patients with a central corneal thickness of 480–499, 500–529, or 530–560 μm. The authors reported a stable posterior corneal elevation regardless of baseline corneal thickness. At the same time, RST correlated with changes in posterior elevation among patients with a baseline corneal thickness of 480–499 μm [19].

Among the ways to improve outcomes of SMILE in thin cornea is the use of corneal crosslinking. Sánchez-González et al. (2021) reported that crosslinking partially preserved good outcomes of surgery [16]. 

To our knowledge, the current study is the first to report an analysis of corneal thickness and procedure on posterior cornea curvature. We found that patients with a baseline corneal thickness of ≤500 μm experienced slower visual acuity recovery that did not impact the overall efficacy and safety of surgery. The results of regression analysis in patients with thin but not in patients with normal corneas show that an increase in the volume of removed tissue leads to a nonlinear increase in Rf back and Rs back. An increase in cap thickness in patients with thin corneas would change Rs back and, possibly, the posterior corneal elevation. At the same time, a smaller cap thickness would leads to better postoperative BCVA values. Jun et al. (2021) found that a cap thickness of 140 μm leads to more changes in corneal biomechanics than a cap thickness of 120 μm, with no significant difference in the efficacy or the predictability of results. However, a higher cap thickness was associated with higher rates of HOA [20].

Insights from our study indicate that there are substantial differences in posterior cornea reaction on SMILE. Therefore, nomogram calculation for patients with normal and thin corneas should also be different. While nowadays it is a standard practice to use RST for planning refractive intervention, it should be noted that this parameter refers to the bidimensional structure of the cornea. However, it may be important to plan interventions and assess risk of complications based on three-dimensional variables, i.e., volume instead of thickness could be more appropriate. For example, although of limited application in SMILE and LASIK, among the planning milestones that can be used is the percent of tissue altered (PTA), which should not exceed 40% to minimize the risk of ectasia. 

Based on correlations between cap thickness and change in posterior cornea curvature, we suggest considering the inclusion of cap thickness as an important variable for the SMILE procedure. Further studies are needed for thorough investigation and discussion regarding relationship between cap thickness and SMILE outcomes depending on baseline corneal thickness.

We plan to expand the follow-up data to evaluate the long-term safety of SMILE in patients with different corneal thicknesses.

Our study has several limitations. Due to the retrospective nature of this study, we had limited follow-up data. Moreover, there were significantly more patients with normal corneas compared to patients with thin corneas from baseline. We did not make attempts at data imputation for this study. The other limitation of this retrospective study is that it was not possible to check the correctness of data entry into source medical documents. Corneal biomechanics are not routinely examination in patients undergoing SMILE within study centers, thus it was not evaluated. At the same time, all patients were followed up to assess the risk of ectasia.

## 5. Conclusions

SMILE appears to be as effective and safe in patients with a corneal thickness of 477–500 μm as in patients with a normal corneal thickness of >500 μm based on a relatively short-term (6 months) follow up. Our study results indicate that cap thickness has a different impact on posterior cornea curvature in patients with different corneal thicknesses. Further studies are needed for the evaluation of the impact of thin corneas on SMILE outcomes and necessity of changing the nomogram. It may also be of importance to consider three-dimensional parameters for HOA risk evaluation and SMILE planning in addition to bidimensional characteristics (i.e., RST). 

## Figures and Tables

**Figure 1 jcm-11-04162-f001:**
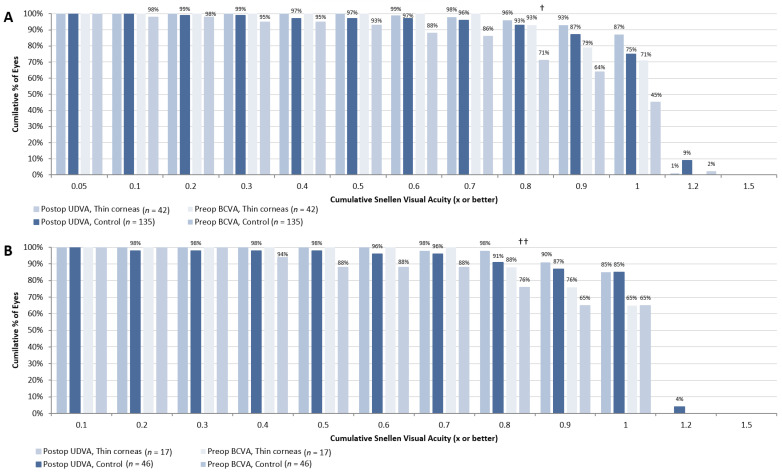
Preoperative corrected distance visual acuity and postoperative uncorrected distance visual acuity at 1 (**A**) and 6 (**B**) months postop. Notes: *n*—number of eyes in each group for time point; UDVA—uncorrected distance visual acuity; BCVA—best corrected visual acuity; †—UDVA intergroup difference significant at *p* < 0.01; ††—UDVA intergroup difference significant at *p* = 0.03.

**Figure 2 jcm-11-04162-f002:**
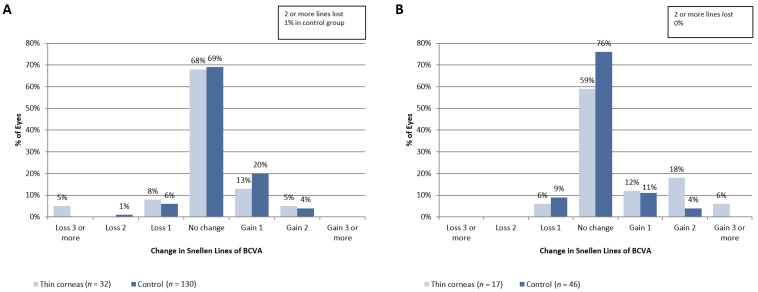
Change in BCVA at 3 months (**A**) and 6 months (**B**) after the procedure compared to preoperative BCVA. Notes: A—evaluation at 3 months after surgery; B—evaluation at 6 months after surgery; BCVA—corrected distance visual acuity; visual acuity was measured by Snellen lines; *n*—number of eyes in each group for time point; no significant intergroup differences observed at 3 or 6 months post-surgery.

**Figure 3 jcm-11-04162-f003:**
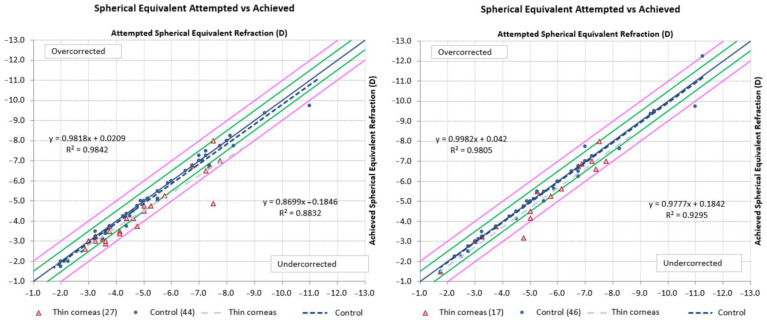
Correlation of attempted and achieved spherical equivalent depending on baseline cornea thickness at 3 months (**left**) and 6 months (**right**). Notes: analysis was performed for 27 and 44 eyes at 3 months, and 17 and 46 eyes at 6 months, for the “thin corneas” and control groups, respectively. Green lines represent 95% confidence interval, pink lines indicate 95% prediction interval.

**Figure 4 jcm-11-04162-f004:**
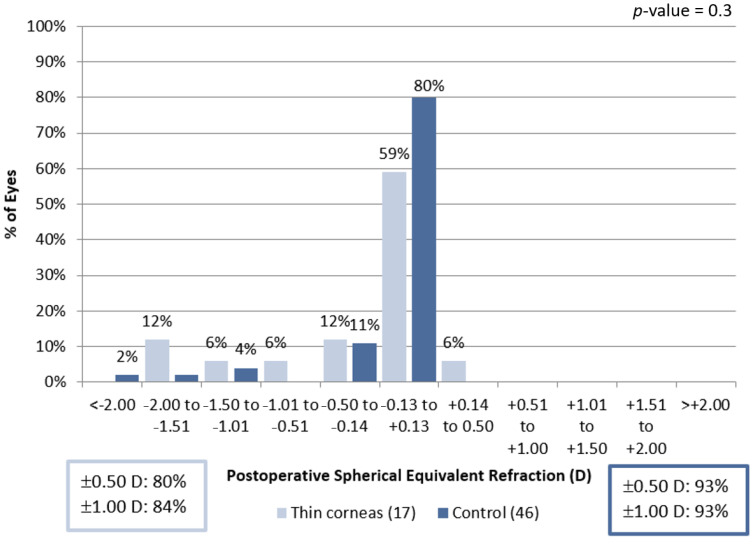
Spherical equivalent refractive accuracy at 6 months.

**Figure 5 jcm-11-04162-f005:**
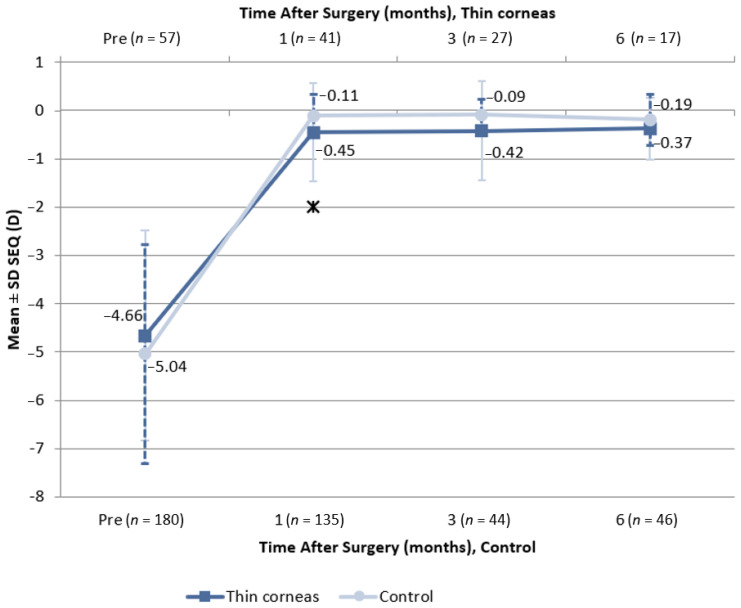
Spherical equivalent stability depending on baseline cornea thickness. Notes: SEQ—spherical equivalent; data are the means and standard deviations (SD); *n*—number of eyes; * – *p*-value < 0.05.

**Table 1 jcm-11-04162-t001:** Baseline patient characteristics.

Variable	Control Group (*n*/*N* = 90/180)	Thin Corneas Group (*n*/*N* = 29/57)
Age, years, M ± SD	29.71 ± 6.91	31.47 ± 7.21
Sphere, D, M ± SD (range)	−4.76 ± 2.17 (−11–−1.25)	−4.41 ± 2.14 (−9.5–−1.5)
Cylinder, D, M ± SD	−0.60 ± 0.68	−0.56 ± 0.55
SEQ (treatment plan), D, M ± SD	−5.0 ± 2.09	−4.57 ± 1.93
BCVA, M ± SD	0.98 ± 0.08	0.96 ± 0.08
OZ, mm, M ± SD	6.55 ± 0.33	6.11 ± 0.64 ^††^
Pachymetry, μm, M ± SD	543.06 ± 30.99	487.77 ± 9.01 ^††^
RST, μm, M ± SD	330.34 ± 40.4	296.63 ± 7.9 ^††^
D-cap, mm, M ± SD	7.57 ± 0.32	7.2 ± 0.7
Lenticule thickness, μm, M ± SD	99.2 ± 24.63	79.6 ± 10.58 ^††^
Volume of removed tissue, mm^3^, M ± SD	3.35 ± 0.57	2.35 ± 0.88 ^††^
Target refraction, D, M ± SD (range)	−0.09 ± 0.39 (−3.5–0)	−0.2 ± 0.48 (−2–0) ^†^

Notes: *n*/*N*—number of patients/eyes; M—mean, SD—standard deviation; SEQ—spherical equivalent, BCVA—best corrected visual acuity; OZ—optical zone size; RST—residual stromal thickness; D-cap—cap diameter; ^†^—significant intergroup difference with *p* < 0.05; ^††^—significant intergroup difference with *p* < 0.01.

**Table 2 jcm-11-04162-t002:** Published studies of small-incision lenticule extraction outcomes in patients with thin corneas.

Author (Year)	Corneal Thickness, μm		Key Results
	Thin Cornea	Control	
Kabakci et al. (2020) [17]	<500, subgroups: 470–485 485–500	–	SMILE was effective regardless of corneal thickness. Rates of HOA tend to increase in the postoperative period.
Zhao et al. (2022) [19]	480–499 500–529	530–560	Efficacy did not differ between groups. Posterior corneal elevation remained stable regardless of baseline corneal thickness. In patients with thin corneas (480–499 μm), residual stromal thickness negatively correlated with changes in posterior elevation.
Ganesh et al. (2015) [18]	<450	–	SMILE was effective in thin corneas, with stable functional outcome and topography. HOA rates did not change significantly over a 12 month observation.
Sánchez-González et al. (2021) [16]	–	–	Preventive corneal crosslinking as an adjunct to SMILE was partially effective for procedure outcome improvement.
Jun et al. (2021) [20]	–	–	A smaller cap thickness (120 vs. 140 μm) was associated with lower rates of HOA. With a cap thickness of 140 μm, greater changes in corneal biomechanics were observed.

Notes: HOA—higher-order aberration.

## Data Availability

No new data were created or analyzed in this retrospective study. Data sharing is not applicable to this article.

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
