# Peer review of "Visual Outcomes of Small-Incision Lenticule Extraction (SMILE) in Thin Corneas"

_jcm, 2022, doi:10.3390/jcm11144162_

Round 1

Reviewer 1 Report

Except for the limitations introduced by patients lost to follow up I think you did a decent job with the data and provide additional information on SMILE with thin corneas. I do think there is room for a future paper if you can get long term follow up especially on the thin cornea subjects as this was too short a time period to detect ectasia.

Overall the paper is clearly understandable but there are some missing articles (in the English sense on 'a', 'the',etc.) and a few misused propositions. I have provided suggestions/corrections for these. A few figures could be improved with some better labeling of axes for which I also made suggestions. I made a comment on the statistics with respect to the treatment of eyes vs patients really as independent.  At least think about the issue but I do not view it as enormously important but some of my colleagues worry about this. 

Author Response

We are very thankful for your comments. Please find attached revised manuscript, all changes in "track change" mode. We also transferred all comments from the pdf file and left replies. Please note that figures will be further changed to comply with JRS/JRCS 6-figure standard according to the comments from other reviewers.

Reviewer 2 Report

The authors aimed to find out whether the thin normal or moderate normal corneal thickness would impact efficacy and safety of SMILE and they found that thin cornea might requite different approach to nomogram calculation.

The topic of this manuscript is very interesting for the refractive laser surgeon and  have a practical a daily implementation. Some issues must be raised prior to continue with the publication process.

Change the title of the manuscript to make it more scientific soundness and include some of the findings in this title in order to catch the audience.

The term thin normal and moderate normal must be changed for another that represent better the thin corneas and normal ones.

Introduce the standards 6 or 9 graph of dan Reinstein in the manuscript. You must follow the official visual and refractive outcomes standard in refractive surgery report.

Update references about SMILE, in a quick search on Pubmed I could find the following, please cite them:

https://pubmed.ncbi.nlm.nih.gov/34259803/

https://pubmed.ncbi.nlm.nih.gov/34937212/

https://pubmed.ncbi.nlm.nih.gov/26221538/

https://pubmed.ncbi.nlm.nih.gov/31119423/

Try to find another updated reference for number 5

The final disclosure od the manuscript is missing

Start the introduction directly with the smile, do not introduce the technique is well-known for refractive surgeons.

Do not include the Sirius paramerts, the belin and ambrosio one is better

Biomechanics was not study in this research this is a major limitation

Do not include question on the introduction

Do no express the info with bullets on method, write then in a sentence.

Figure should be all redone according standard reporting graph of JRS o JCRS

In  discussion a summary table of previous thin cornea with smile study results could find interesting

Include future research lines at the end of the discussion

The PTA has been demonstrated as limited application, do not include as a necessary in the discussion or conclusion section.

Author Response

We very much appreciate those profound comments you gave to our manuscript. Please find it attached. We did our best to pay attention to all points considered. Below we shortly describe changes done in the manuscript that you can find attached for further review.  1. Manuscript title revised to add scientific soundness and reflection of study results. 2. All graphs changed to comply with JRS standard figures. 3. References you mentioned added within text, reference 5 updated. 4. Introduction changed to a more direct type. 5. Methods section reworked to a plain text. 6. Discussion improved, we added a supportive table with SMILE studies' results, limitations also expanded according to the comments.  Considering use of Belin\Ambrosio parameters, centers that participated in this study use Sirius unit as a part of routine practice. For that reason we were able to add only Sirius parameters. Considering wide use of this equipment, we think our data would be relevant for clinicians.  The First edition of the manuscript contained PTA as a point for discussion over planning and risk assessment in SMILE. Our goal was to pay attention to the use of bidimensional characteristics such as RST for evaluation of event risk in a three-dimensional model. We would like to keep these highlights and therefore revised this part of the discussion section.  Once again thank you for the comments on our works. We are looking forward to further discussion.

Round 2

Reviewer 2 Report

Comments solved

Author Response

Thank you once again for thorough review of our paper. We will further work on it's improvement.